# Dynamic Impacts of External Uncertainties on the Stability of the Food Supply Chain: Evidence from China

**DOI:** 10.3390/foods11172552

**Published:** 2022-08-23

**Authors:** Jingdong Li, Zhouying Song

**Affiliations:** 1Institute of Geographic Sciences and Natural Resources Research, Chinese Academy of Sciences, Beijing 100101, China; 2Key Laboratory of Regional Sustainable Development Modeling, Chinese Academy of Sciences, Beijing 100101, China

**Keywords:** food supply chain, external uncertainties, dynamic impact, risk identification, three-dimensional impulse response

## Abstract

The food supply chain operates in a complex and dynamic external environment, and the external uncertainties from natural and socio-economic environment pose great challenges to the development of the food industry. In particular, the COVID-19 pandemic and Russia–Ukraine conflict have further exacerbated the vulnerability of the global food supply chain. Analyzing the dynamic impacts of external uncertainties on the stability of food supply chain is central to guaranteeing the sustainable security of food supply. Based on the division of food supply chain and the classification of external uncertainties, the TVP-FAVAR-SV model was constructed to explore the dynamic impacts of external uncertainties on food supply chain. It was found that the impacts of external uncertainty elements were significantly different, the combination of different external uncertainty elements aggravated or reduced the risks of food supply chain. And some uncertainty elements had both positive and negative impacts in the whole sample period, as the magnitude and direction of the impacts of various uncertainties in different periods had time-varying characteristics.

## 1. Introduction

Food security is the guarantee for human survival, economic development, and social harmony. The outbreak of the global food crisis in the early 1970s and in 2008 caused food insecurity in many countries and induced great tension in the world [1]. Recently, the continuation of the COVID-19 pandemic and the outbreak of military conflict between Russia and Ukraine have seriously threatened global food security, and the factors affecting food insecurity have also received extensive attention. Food security is a dynamic condition resulting from the interaction of various factors [2]. Stability is a significant indicator to measure food security [3,4], which describes the extent of impacts on the food supply chain when disrupted, such as market fluctuations, extreme weather, conflicts, political crises, policy changes, epidemics, and natural disasters [5,6]. And a resilient food supply chain will have higher stability [7,8,9]. The stability of the supply chain is affected by internal structures and external impacts [10], which is the combination of supply chain management and environmental management [11]. As an important part of supply chain management and an extension of environmental management, risk management is used to evaluate internal operational risks and external environmental risks [12,13]. Internal operational risks are mainly caused by the instability of supply chain configuration conditions such as production technology, management experience, information technology, business strategy, and contractual relationships [5,14,15,16,17]. External environmental risks include the impacts of natural environment and socio-economic environment, which affect the food supply chain by impacting the production conditions and production processes [18,19].

Studies on the food supply chain from different perspectives has played an important role in promoting the development of food supply chain. Based on the general supply chain theory, Ouden et al. [20] found a way out of previous research dilemma of simply identifying production and processing as food safety, through the integrated analysis of all sectors from production to consumption, to put forward the concept of food supply chain. Some viewpoints hold that the food supply chain is a strategic interaction between the government, manufacturers, and farmers that is nurtured by endogenous consumer demand [21]. From the components of food supply chain, it is mainly composed of production sector, circulation sector, and consumption sector [22,23]. Meanwhile, some main factors promoting the development of food supply chain, such as quality, technology, logistics, information technology, regulatory framework, and consumers, are vulnerable to the external environment [24,25,26]. When the deterioration of the external environment causes changes of these key factors, the stability of food supply chain is seriously challenged [27], and it is crucial to evaluate and manage these supply chain risks [28]. Risk refers to the loss caused by the deviation of the actual situation from the expected situation under given conditions and specific periods [29]. Compared with risk, when the magnitude and direction of an event’s impact on the supply chain are temporarily unknown, it is called uncertainty [30]. Alternatively, risk usually produces negative effects, while uncertainty produces both positive and negative results [30,31].

The food supply chain usually faces complex challenges from external economic, social and natural environment when maintaining stability [13,32]. Thus, it is imperative to evaluate the economic, social, and natural environment in the supply chain [13,32,33]. In particular, the COVID-19 pandemic has made organizations and scholars realize that they should go beyond traditional social and environmental concerns and turn to the compound impacts of the environment, health, and economic society to study the disruption and resilience of supply chain [34,35]. Firstly, changes in natural environment have impacts on food supply chain [22]. Climate change has changed the temporal-spatial distribution pattern of water resources and increased the pressure on global agricultural water use [36]. Among them, global warming increased the probability of hydrological extreme events [37], which reduced crop yield, destroyed farmland ecosystem [38], and influenced animal health and livestock processing industry [39]. In recent years, the global outbreak of highly pathogenic avian influenza disrupted farm operation, international trade and household consumption [40]. The economic recession and food supply chain interruption caused by COVID-19 in 2020 had a great impact on the global food system, poverty, and nutrition [41], and led to serious food insecurity for more than 40 million people in 17 countries [6]. Secondly, socio-economic uncertainties such as economic crisis, energy-related fluctuations, political instabilities, regional conflicts, and unstable policies can lead to the increase of production costs, decline of production efficiency, rise of sales prices, decrease of inventories, vulnerability of trade networks, reduction of family income, and change of consumption expectations and dietary habits, which have great impacts on the whole food supply chain [1,42,43,44,45,46,47].

China is the largest grain producer and importer. Limited by the natural environment, China has to ensure food and nutrition security of 18% of the world’s population with only 9% of the world’s cultivated land and 6% of the world’s water resources [48]. China also has frequent occurrence of various natural disasters. Events such as flood, drought, hail, earthquake, debris flow, and extreme weather have seriously affected its agricultural production [49]. Moreover, China is in an important period of social transformation. The contradiction between economic development and benefit distribution leads to the continuous emergence of social public events, which aggravates people’s sense of crisis [50]. However, China’s food industry is widely scattered. Agri-product processing and food manufacturing enterprises are mainly small and medium-sized [51], with little awareness of risk prevention and weak resistance to risks [52]. Food supply chain operates in a complex and dynamic external environment. Not only can the external uncertainties affect a particular sector in the food supply chain, but also the impacts can transfer among sectors [1,32]. The combination of different external uncertainty elements also has significantly different impacts on the food supply chain [22,34,35].

Research on the impacts of external uncertainties (or risks) on food supply chain is of great significance to eliminate global hunger and achieve sustainable security of food supply [53]. The research on the definition, risk identification, and classification of food supply chain is relatively mature [21,22,23]. However, many existing studies only regard natural disasters or emergencies as the external risks of the food supply chain, but ignore the impacts of socio-economic environment and the superposition impacts of the interaction between natural and socio-economic environment on the food supply chain. The complex and dynamic external environment, especially the uncertainties from socio-economic and natural environment, pose great challenges to the development of agricultural industry and food industry [13,32]. Therefore, the study of food supply chain should more comprehensively analyze the impacts of various external uncertainties on the food supply chain, and grasp the interaction between the interdependent links [2]. Meanwhile, considering the high frequency, time-varying quality, complexity, and uncertainty of external environmental elements, many analysis methods in the existing research (like event analysis method, economic policy uncertainty index method, interpretive structural modeling, local projections, simultaneous equations, global computable general equity model, and time-varying parameter vector autoregression model) cannot effectively match these characteristics [41,54,55,56,57,58,59].

To improve these deficiencies, this paper first divides the main sectors of the food supply chain and selects a large number of indicators from the two aspects of natural and socio-economic environment to measure the external uncertainties, which serves as entry points for the study. Then, the TVP-FAVAR-SV model is constructed, and the common factors reflecting the external economic environment uncertainties are extracted through latent factor extraction method. With the help of the three-dimensional impulse response of the representative variables in the sectors of the food supply chain to all external uncertainty elements, this paper studies the dynamic impacts of a single external uncertainty element on the food supply chain, and the different impacts of the combination of multiple external uncertainty elements on the food supply chain. Compared with existing research, the three-dimensional impulse responses based on the TVP-FAVAR-SV model can depict the dynamic and complex impacts of external uncertainties more clearly. The impulse response results show the time-varying characteristics of the uncertainties’ impacts, and reveal that the combination and transmission of various uncertainties’ impacts are the main reason for aggravating or reducing the food supply chain risks.

## 2. Materials and Methods

In this section, first, we divide the major sectors of food supply chain and classify the external uncertainties, which serve as entry points for the study. Second, the impact mechanisms of external uncertainties on all sectors of the food supply chain are identified. Finally, the construction of empirical model and data sources are introduced.

### 2.1. Major Sectors of Food Supply Chain

The food supply chain comprises multiple sectors and sub-elements, and each sub-element is independent but closely linked [60]. The synergistic and coordinated interactions between different sectors are important to maintain the stability of the food supply chain [32]. The food supply chain can be divided into primary agri-products supply sector, food production sector, circulation and trade sector, and consumption sector [2,22,23], as shown in Figure 1. Among the main sectors of the food supply chain, the primary agri-products supply sector mainly refers to the process in which primary agri-products are provided to food enterprises as raw materials through domestic production and imports. The production sector involves a process in which food enterprises process primary agri-products (agri-product processing industry) or manufacture food products (food manufacturing industry) to make profits by selling finished products. The circulation and trade sector contains the process of food circulation in the domestic and international markets through wholesalers, retailers, importers, exporters, and other subjects. Lastly, the consumption sector mainly refers to the process in which consumers buy food according to their own needs, purchasing power, and consumption expectations.

### 2.2. Sources and Classification of External Uncertainties

The external uncertainties include economic uncertainties, social and natural public emergencies [1,19,22]. This paper draws on the research of Bernanke et al. [61] and Liu et al. [62] for measuring economic uncertainties and selects a large number of macroeconomic indicators to reflect the complexity and uncertainty of China’s economy. Specifically, 67 indicators are selected to analyze China’s economic situation, from the aspects of industrial development, social fixed investment, currency and financing, interest rate, exchange rate, consumption, trade cooperation, securities market, government finance, prosperity index, and price index, so as to better describe the uncertainties of China’s macro-economy environment (Figure 2). Selection of social and natural public emergencies is based on the definition of public emergencies in the Emergency Response Law of the People’s Republic of China promulgated by the Chinese government, and public emergencies included natural disasters, accident disasters, public health emergencies, and social security events [50,63]. In light of the increasing impacts of public health events on food security in recent years, this paper takes public health events as a separate index. Finally, this paper divides social and natural public emergencies into public health emergencies and disaster emergencies. Combining with the types of Statutory Infectious Diseases issued by the National Health Commission of the People’s Republic of China, 36 infectious diseases such as COVID-19, SARS, and H5N1 are selected to represent public health events, as shown in Figure 2. Disaster emergencies include natural disasters, accident disasters, and social security events (Figure 2).

### 2.3. Impacts on Identification and Representative Indicators Selection

#### 2.3.1. Primary Agri-Products Supply Sector

The impacts caused by the external uncertainties on the primary agri-products supply sector are mainly reflected in labor force reduction, disaster area increase, agricultural production input increase, variety selection, premature harvesting, policy changes, and subsidy shortage [22,43,57,64,65], which lead to the supply instability, price volatility, and trade fluctuations of primary agri-products (Figure 3). Therefore, the purchase price, production value, and trade volume of primary agri-products are selected as representative indicators to analyze the dynamic impacts of external uncertainties on the supply of raw materials.

#### 2.3.2. Food Production Sector

The impacts caused by the external uncertainties on the food production sector are mainly reflected in the impacts on the cost, inventory, production strategy, support services and financial risks of food production enterprises (including agri-products processing enterprises and food manufacturing enterprises) [66,67,68], which lead to the instability of food supply, price fluctuation and decrease of profits for enterprises. The risks in this sector can be divided into two categories according to the source: one is the direct impact of the external uncertainties on the food production sector, and the other is the transmission of impacts of the external uncertainties on the primary agri-products supply sector (Figure 3). The sales price of food enterprise, production value, and profit margin of the food industry are selected as the representative indicators for the production sector to analyze the impacts of public health emergencies, disaster emergencies, and economic uncertainties on the production sector.

#### 2.3.3. Circulation and Trade Sector

The impacts caused by the external uncertainties on circulation and trade sector are mainly reflected in the impacts on food supply, operation costs, market selection, market access, market sanctions, and trade agreements in the food market, which lead to price volatility, consumption fluctuation and trade fluctuation [1,22,69,70]. The sources of risks in this sector can also be divided into two categories: one is from the transmission of impacts of external uncertainties on food production sector and consumption sector, and the other is the direct impact of external uncertainties on food circulation and trade sector (Figure 3). Food retail, food products export, and food products import are selected as representative indicators to analyze the impacts of external uncertainties on the circulation and trade sector.

#### 2.3.4. Consumption Sector

In the consumption sector, the decision-making behavior of consumers on food purchase is the main content. The impacts of external uncertainties on consumption are mainly realized by changing consumers’ purchasing power, consumption habits, and expectations (Figure 3). In the face of public emergencies, consumers will adjust food consumption expectations in the short term and increase food reserve to ensure future living, and it is more prominent among risk-averse consumers [15]. Meanwhile, the rise in food prices caused by natural disasters also has impact on consumers’ purchasing power and consumption habits [70]. Factors related to consumer income in economic uncertainties can affect food consumption ability in the short term [71]; factors related to market development can change consumers’ consumption expectations through demonstration effects in the long term [8,72]. Meanwhile, with the increase of consumers’ income, consumption habits will be rapidly upgraded from plant-based food to animal-based food [73]. Therefore, this paper selects the food consumption value and food consumption price as the representative indicators of the consumption sector.

### 2.4. TVP-FAVAR-SV Model

The external uncertainties to be investigated in this paper cover many indicators, especially 67 economic indicators are used to measure the economic uncertainty, and which cannot be fully included for analysis. In order to show as many of the impacts of external uncertainties as possible, it is necessary to be more prepared to grasp the time-varying characteristics of various impacts, and to improve the ability to explain the relationship among the variables in the system, so it is crucial to analyze with the help of an appropriate model. Considering that the characteristics of some indicators can be described by the common part, the dimensionality reduction of indicators can be achieved by extracting the common part [61]. Kazi et al. [74] and Koop et al. [75] constructed the time-varying parameters factor-augmented vector autoregression with stochastic volatility (TVP-FAVAR-SV) model to realize time-varying characteristic analysis with random fluctuation based on factor broadening. Therefore, this paper constructs the TVP-FAVAR-SV model to analyze the dynamic impacts of external uncertainties on China’s food supply chain by reducing the dimension of economic indicators.

In order to study the time-varying quality of the parameters, the model needs to be extended to a time-varying parameter form. The basic expression of the time-varying parameters factor-augmented VAR (TVP-FAVAR) model takes the form:(1)$$\left[\begin{array}{c} F_{t}\\ Y_{t} \end{array}\right]=\psi_{1,t}\left[\begin{array}{c} F_{t-1}\\ Y_{t-1} \end{array}\right]+\psi_{2,t}\left[\begin{array}{c} F_{t-2}\\ Y_{t-2} \end{array}\right]+\cdots +\psi_{p,t}\left[\begin{array}{c} F_{t-p}\\ Y_{t-p} \end{array}\right]+\upsilon_{t}$$
where $\psi_{i,t}(i=1,\cdots ,p;t=1,\cdots ,T)$ is a $\left(m\times m\right)$ matrix of time-varying coefficients; $\upsilon_{t}\sim N(0,\Omega_{t})$, $\Omega_{t}$ is a $\left(m\times m\right)$ time-varying covariance matrix, m=k+l.

According to the methods of Primiceri [76] and Nakajima [77] in dealing with the covariance matrix, $\Omega_{t}$ is decomposed as:(2)$$A_{t}\Omega_{t}{A^{\prime }}_{t}=\Sigma_{t}{\Sigma^{\prime }}_{t},\;\Omega_{t}=A_{t}^{-1}\Sigma_{t}{\Sigma^{\prime }}_{t}({A^{\prime }}_{t}^{-1})$$
where $\Sigma_{t}$ is a $\left(m\times m\right)$ diagonal matrix, and $A_{t}$ is a $\left(m\times m\right)$ lower-triangular matrix with the diagonal elements equal to 1.
(3)$$\Sigma_{t}=\left[\begin{array}{cccc} \sigma_{1,t} & 0 & \cdots & 0\\ 0 & \ddots & \ddots & \vdots \\ \vdots & \ddots & \ddots & 0\\ 0 & \cdots & 0 & \sigma_{m,t} \end{array}\right],\;A_{t}=\left[\begin{array}{cccc} 1 & 0 & \cdots & 0\\ a_{21,t} & 1 & \ddots & \vdots \\ \vdots & \ddots & \ddots & 0\\ a_{m1,t} & \cdots & a_{m(m-1),t} & 1 \end{array}\right]$$

Stacking the row vector of $\psi_{i,t}$ in Equation (1), we define:(4)$$\psi_{t}=(vec(\psi_{1,t}{)}^{\prime },\cdots vec(\psi_{p,t}{)}^{\prime })$$
(5)$$\log\sigma_{t}=(\log{\sigma^{\prime }}_{1,t},\cdots ,\log{\sigma^{\prime }}_{m,t})$$
(6)$$\alpha_{t}=({a^{\prime }}_{j1,t},\cdots ,{a^{\prime }}_{j(j-1),t})$$

Following the assumptions as Koop et al. [78], for each time period, the random walk evolutions of $\Lambda_{t}$, $H_{t}$, $\psi_{t}$, $\alpha_{t}$ and $\log\sigma_{t}$ are assumed to be a mixture of two normal components, and the random walk evolutions can be expressed as:(7)$$\Lambda_{t}=\Lambda_{t-1}+J_{t}^{\Lambda }\eta_{t}^{\Lambda },\;H_{t}=H_{t-1}+J_{t}^{H}\eta_{t}^{H},\;\psi_{t}=\psi_{t-1}+J_{t}^{\psi }\eta_{t}^{\psi }$$
(8)$$\alpha_{t}=\alpha_{t-1}+J_{t}^{\alpha }\eta_{t}^{\alpha },\;\log\sigma_{t}=\log\sigma_{t-1}+J_{t}^{\sigma }\eta_{t}^{\sigma }$$
where, $\eta_{t}^{\theta }\sim N(0,\Sigma_{\theta })$ are independent innovation vectors, $\theta_{t}\in \left\{\Lambda_{t},H_{t},\psi_{t},\alpha_{t},\log\sigma_{t}\right\}$, $\Sigma_{\theta }$ are innovation covariance matrices corresponding to $\Lambda_{t}$, $H_{t}$, $\psi_{t}$, $\alpha_{t}$ and $\log\sigma_{t}$; $J_{t}^{\theta }$ are random variables, and its different values lead to structural breaks of the respective innovation errors in time-varying parameters, $J_{t}^{\theta }=1$ mean time-varying parameters model, $J_{t}^{\theta }=0$ mean constant parameters model.

Thus, Equation (1) can be rewritten as:(9)$$G_{t}=\Lambda_{t}Z_{t}+W_{t}\mu_{t}^{G}$$
(10)$$Z_{t}=\psi_{1,t}Z_{t-1}+\psi_{2,t}Z_{t-2}+\cdots +\psi_{p,t}Z_{t-p}+A_{t}^{-1}\Sigma_{t}\mu_{t}^{Z}$$
where
(11)$$G_{t}^{\prime }=\left[X_{t}^{\prime },Y_{t}^{\prime }\right],\;Z_{t}^{\prime }=\left[F_{t}^{\prime },Y_{t}^{\prime }\right]$$
(12)$$W_{t}=diag(\exp(h_{1,t}/2),\cdots ,\exp(h_{n,t}/2),\cdots ,\exp(H_{n,t}/2),0_{1\times l})$$
(13)$$W_{t}W_{t}^{\prime }={\left[H_{t},0_{1\times l}\right]}^{\prime },\;\Lambda_{t}=\left[\begin{array}{cc} \Lambda_{t}^{f} & \Lambda_{t}^{y}\\ 0_{l\times k} & I_{1\times l} \end{array}\right]$$
where random interference term $\mu_{t}^{G}$ and $\mu_{t}^{Z}$ follow the standard normal distribution and independent with each other.

Inserting Equation (10) into (9) can obtain the final TVP-FAVAR-SV form:(14)$$G_{t}=\Lambda_{t}\psi_{1,t}Z_{t-1}+\Lambda_{t}\psi_{2,t}Z_{t-2}+\cdots +\Lambda_{t}\psi_{p,t}Z_{t-p}+\xi_{t}$$
(15)$$\xi_{t}=\Lambda_{t}(A_{t}^{-1}\Sigma_{t})\mu_{t}^{Z}+W_{t}\mu_{t}^{G}$$

### 2.5. Data Sources

Monthly data are selected in this paper to describe the external uncertainties as carefully as possible and maintain the consistency of data frequency. Considering data availability, the sample interval selected in this paper is from March 2005 to June 2020. In primary agri-products supply sector, primary agri-products trade includes vegetable products and animal products, and the specific product types and corresponding HS codes are shown in Table A1 (Appendix A). In the production sector, the food industry is divided into agri-products processing industry and food manufacturing industry, according to the standards of China National Bureau of Statistics, and the specific industrial classification is shown in Table A1 (Appendix A). In the circulation and trade sector, the trade types and corresponding HS codes are shown in Table A1 (Appendix A). The data used in this paper are obtained from the Chinese National Bureau of Statistics, China Customs Database, Wind Database, EM-DAT Database, and National Health Commission of the People’s Republic of China, and the descriptive statistics of all variables are shown in Table 1. Before the empirical analysis, year-over-year processing on all variables is conducted, in which the monthly data of the year t is divided by that of the corresponding month of the year t-1 to obtain the year-over-year data of the variable (variables that are year-over-year data themselves are not processed), to eliminate the impacts of inflation on price and the seasonal factors of monthly time series. Finally, to facilitate the empirical analysis of the TVP-FAVAR-SV model, all series are standardized and transformed into a standard sequence with a mean value of 0 and standard deviation of 1. The empirical operations in this paper are implemented by the MATLAB R2018b (MathWorks, Natick, MA, USA).

## 3. Empirical Results

### 3.1. Identification of Common Factors of External Economic Uncertainties

This paper draws on the latent factor extraction method proposed by Bernanke et al. [61] to extract common factors from high-dimension economic indicators, to realize dimension reduction of indicators. With the help of sensitivity analysis, the number of common factors to be extracted is determined by gradually increasing its number [79]. When the number of extracted common factors is 4, increasing the number of common factors does not have a substantial impact on the results of the model, and the results of common factors are shown in Figure 4. Comparisons are made about the trend of four common factors and that of the original 67 indicators. Variables with similar trend are grouped together, and finally four groups of graphs are obtained, as shown in Figure 4a–d.

In Figure 4a, the overall trend of common factor 1 is similar to that of deposit interest rate, loan interest rate, foreign exchange reserve, price index, etc., so it can be named interest rate uncertainty (IRU) factor. In Figure 4b, the overall trend of common factor 2 is similar to that of social financing, currency supply, market value of listed companies, stock trading value, fund trading value, futures trading value, etc., so it can be named financial uncertainty (FU) factor. In Figure 4c, the overall trend of common factor 3 is similar to that of industrial added value, government finance, fixed asset investment, domestic credit, import and export index, prosperity index, etc., so it can be named social development uncertainty (SDU) factor. In Figure 4d, the overall trend of common factor 4 is similar to that of consumption index, exchange rate level, etc., so it can be named consumption uncertainty (CU) factor. The paper further uses the general factor analysis to test the results of common factors (obtained by the latent factor method) and the validity of the naming method. The results of the general factor analysis are shown in Table A2 (Appendix A). From the results of general factor analysis, the classification of economic indicators is similar to the results of the latent factor method, which verifies the effectiveness of the naming method for the common factors obtained by the latent factor method.

### 3.2. Stationary Test

Since this paper studies the relationship between time series, before empirical analysis, this paper uses unit root test to examine the stationarity of all variables in the TVP-FAVAR-SV model, in order to eliminate the influence of random law differences in non-stationary series [80,81]. The unit root test results of each time series are shown in Table 2. According to the *p*-value of ADF-statistic, all variables have passed the unit root test.

### 3.3. Impulse Response Result Analysis

Based on the common factor extraction and unit root test, this paper further explores the three-dimensional impulse responses of representative variables in each sector of food supply chain to positive shocks of public health emergencies (PHE), disaster emergencies (DE), interest rate uncertainty factor (IRU), financial uncertainty factor (FU), social development uncertainty factor (SDU), and consumption uncertainty factor (CU).

#### 3.3.1. The Impulse Responses of Primary Agri-Products Supply Sector

The impulse responses of primary agri-products purchase price (PAPP)

Figure 5 shows in the three-dimensional impulse responses of PAPP to the positive shocks of external uncertainties from March 2005 to June 2020. In Figure 5, the x-axis represents the duration of impulse responses of PAPP to positive shocks of external uncertainties, the y-axis represents the occurrence time of positive shocks of external uncertainties, and the z-axis represents the response level of PAPP to positive shocks of various uncertainty elements.

The positive shocks of external uncertainties mainly have positive impacts on PAPP. FU and DE have relatively large impacts, while that of CU is the smallest. The impacts of DE have obvious intermittence.

From October 2009 to September 2010, IRU had great impacts on PAPP, and the Chinese government implemented moderately loose monetary policies to cause an increase in the positive impacts of IRU on PAPP [43,82]. When the liquidity is low (for example, from August 2012 to September 2013), the positive impacts of IRU on PAPP decrease, and can even shift to negative impacts. The results can verify the findings of Tian et al. [83].

From April 2007 to February 2008 and from May 2009 to March 2010, FU had great positive impacts on PAPP, because the agri-products price rose rapidly with the high liquidity of domestic currency and assets in these two periods [43].

From February 2009 to October 2010, the impacts of SDU on PAPP increased remarkably, mainly due to the significant growth of PAPP driven by the ℌfour trillionℍ economic stimulus plan implemented by the Chinese government [43,82]. From August 2018 to June 2020, the Sino-US trade war and the outbreak of COVID-19 had negative impacts on economic development; to avoid drastic fluctuation of agri-products price, the Chinese government strengthened price supervision [8], and consequently, PAPP decreased slightly.

From November 2008 to April 2011, the positive impacts of CU on PAPP were significantly greater than that in other periods. Reasons include the high level of consumer confidence and expectation, and the depreciation of CNY exchange rate. Therefore, strong demand in both domestic and international markets significantly increased PAPP [84].

The continuous heavy rainfall in the Yangtze River Basin from July to August 2010 seriously impacted agricultural production, which led to the increase of PAPP from October 2010 to February 2011, due to the lagging effect of agricultural production on prices. From March 2015 to September 2016, due to severe droughts and floods in North China, Northeast China, and Northwest China, as well as the Yellow River and Yangtze River Basins, PAPP increased significantly during this period.

From March 2010 to August 2010, the positive impacts of PHE on PAPP decreased significantly, because the outbreak of influenza A (H1N1) at the end of 2009 reduced consumption of livestock products and pork [85]. Moreover, the outbreak of COVID-19 in 2020 exceeded other national notifiable diseases in terms of impact time, scope, and death toll, but its impacts on the increase of PAPP were much smaller. Because the pandemic prevention measures and the improved agricultural policies taken by the Chinese government [8,86].

2.The impulse responses of production value of primary agri-products (PVPA)

The positive shocks of external uncertainties have both positive and negative impacts on PVPA, with the greatest impacts being of SDU, followed by DE, and CU has the least impacts (Figure 6).

Before 2006, IRU mainly had negative impacts on PVPA due to the downturn of the primary agricultural product market, and the rise of interest rate further increased the loan pressures of agricultural enterprises, agents, and purchasers [87]. After 2006, with the continuous improvement of agricultural support policies, the enthusiasm of participants gradually increased, and the negative impacts of rising interest rate gradually decreased. Meanwhile, price index component in the IRU can also push up PVPA in the short term [43]. Thus, after 2006, IRU first had positive impacts on PVPA, and then had negative impacts.

FU increases PVPA by improving the financing environment of agricultural enterprises and raising primary agri-products price [88], and has a significant pull-up effect during the period of high liquidity (for example, from May 2009 to March 2010). Meanwhile, FU can also have negative impacts on PVPA by raising production costs and opportunity costs.

Since 2006, the Chinese government has implemented a series of agricultural support policies, which have made great contributions to ensuring agricultural production, narrowing the income gap of farmers and maintaining the number of rural laborers [44,54,89]. Thus, SDU has greater positive impacts on PVPA. Meanwhile, during the period of high liquidity (from May 2009 to March 2010), the positive impacts of SDU increased significantly.

CU has negative impacts on PVPA first, and then positive impacts, but the overall level of impacts is small. With the improvement of residents’ income level, people will increase the consumption of durable goods, high-end goods, or luxury goods, resulting in the negative impacts on PVPA; meanwhile, people’s consumption upgrading can also stimulate agricultural production value [73]; thus, CU also has positive impacts on PVPA.

DE has negative impacts on PVPA first, and then positive impacts. The occurrence of DE, especially natural disasters, has relatively great negative impacts on PVPA in the short term. Meanwhile, due to the support and protection of agricultural policies, primary agri-products will quickly resume production after disasters, so PVPA will experience compensatory increase in the long term [8,90]. Due to the continuous and widespread occurrence of droughts and floods, the rise in agri-product prices can no longer compensate for the losses of PVPA caused by the reduction in quantities, and this resulted in a substantial decline in PVPA from 2015 to 2016.

PHE has negative impacts on PVPA first, and then positive impacts, but both positive and negative impacts are small. It reflects that policies implemented by the Chinese government to support and protect agriculture, as well as the emergency prevention and control measures have effectively reduced the impacts of PHE on PVPA.

3.The impulse responses of primary agri-products export (PAE)

The positive shocks of IRU mainly have negative impacts on PAE, while FU, SDU, and CU, DE, and PHE have both positive and negative impacts. Among them, DE has the greatest impacts on PAE, followed by FU and FU, while CU has few impacts (Figure 7).

The increase of IRU means the increase of loan interest rate or price index. On the one hand, it affects PAE by influencing the borrowing costs of exporters [91]; on the other hand, it affects PAE by influencing production costs and export profits of food enterprises. The recovery of loan interest rate and the substantial increase of the agri-production material price index after 2016, in particular, caused a significant decline in PAE.

FU first has short-term negative impacts, followed by long-term positive impacts. The reason for the short-term negative impacts is that the FU raised the price of agri-products, thus reducing the price competitiveness of its exports. The long-term positive impacts are realized by improving the financing environment of export enterprises [88].

SDU increases PAE by improving quantity and quality of agricultural production, and strengthening interregional agricultural trade consultation [92]. However, the outbreak of the Sino-US trade war in March 2018 led to a significant decline in its positive impacts and an increase of its negative impacts.

CU increases PAE by raising the exchange rate of foreign currency to CNY, and reduces PAE by stimulating consumption when the exchange rates are relatively stable, such as the time from October 2009 to October 2012 [82].

DE and PHE first have great negative impacts on PAE, followed by positive impacts. It indicates that the occurrence of DE and PHE reduce PAE in the short term (for example, the extreme droughts and floods that occurred continuously and in large scales from 2015 to 2016 made these negative impacts even more significant), but when the disaster passes and agriculture production recovers, PAE will experience compensatory growth [8].

4.The impulse responses of primary agri-products import (PAI)

The positive shocks of financial uncertainties, CU and PHE, mainly have positive impacts on PAI. IRU, SDU, and DE have both positive and negative impacts. DE has the greatest impacts on PAI, followed by SDU, and FU has the least impacts (Figure 8). IRU increases PAI by raising domestic agricultural production costs in the short term, and reduces PAI by raising borrowing costs of importing enterprises in the long term [88,91]. Therefore, the IRU first has positive impacts and then negative impacts. FU mainly affects PAI by improving import enterprises’ credit level and financing scale [8,88].

SDU reduces the dependence on agri-products imports by increasing domestic supply and applying relevant trade policies (such as non-tariff barriers, anti-dumping investigations). It is also found that during the implementation of the “four trillion” economic stimulus plan (from February 2009 to October 2010), SDU had significant positive impacts, which reflects the strong orientation of the plan [93]. The impacts of CU on PAI gradually decreased, showing that the role of consumption in PAI increase is weakening.

DE and PHE have significant positive impacts on PAI, and impacts of DE are relatively great. It indicates that DE is still the main factors affecting the PAI compared with the impacts of economic uncertainties.

#### 3.3.2. The Impulse Responses of Food Production Sector

The impulse responses of sales price of food enterprise (SPFE)

The positive shocks of external uncertainties mainly have positive impacts on SPFE, with the greater impacts of FU and SDU, and the smallest of DE (Figure 9). IRU has more prominent positive impacts during periods of high currency and asset liquidity (for example, April 2007 to February 2008 and May 2009 to March 2010). To cope with the negative impacts of the Sino-US trade war and COVID-19, the Chinese government implemented a prudent monetary policy to guarantee moderate liquidity (non-interest rate adjustment) from June 2019 to June 2020. Therefore, the positive shocks of FU also showed great positive impacts during this period.

The positive impacts of SDU on SPFE increased significantly from April 2007 to February 2008 and from May 2009 to March 2010. CU has greater pulling effects on SPFE in time of high liquidity, and the impacts of CU on SPFE decreased with time. The continuous heavy rainfall in the Yangtze River Basin from July to August 2010 led to the unstable supply of primary agri-products, so the impacts of DE on SPFE also increased significantly during this period.

PHE mainly has positive impacts on SPFE, but it had significant negative impacts from January 2010 to October 2010. The outbreak of influenza A (H1N1) from October 2009 to April 2010 led to a significant decrease in the consumption of poultry and pork products [85], which further caused the decline of SPFE. Similarly, from January 2017 to December 2017, the significant decline in the positive impacts of PHE was due to the outbreak of H7N9.

2.The impulse responses of production value of food industry (PVFI)

The positive shocks of external uncertainties have both positive and negative impacts on PVFI, with the greatest impacts of DE, followed by PHE, and IRU having the least impacts (Figure 10). The price index component in the IRU can also push up PVFI in the short term [43], while the rise in interest rate increases the loan costs of food enterprises in the long run, stunting PVFI growth. However, both the positive and negative impacts of the positive shocks of IRU are small.

The positive shocks of FU increase PVFI by improving the food enterprises’ financing environment and raising food price. The impacts of FU on PVFI will increase in time of high liquidity [88]. SDU has positive impacts on PVFI by influencing the policy dividends and development prospects of food enterprises and the impacts are relatively stable [8]. The impacts of CU on PVFI are volatile, but the overall impacts are small, indicating that PVFI is not significantly affected by consumption on the premise of “inelastic demand” for food.

DE and PHE first have great negative impacts on PVFI, followed by positive impacts. It shows that when DE and PHE occur, PVFI will decrease in the short term; after the disaster pass and enterprises resume production, PVFI will experience a compensatory increase [90].

3.The impulse responses of profit margin of food industry (PMFI)

The positive shocks of IRU mainly have negative impacts on PMFI. The positive shocks of SDU mainly have positive impacts. FU, CU, DE, and PHE have both positive and negative effects. DE has the greatest impacts on PMFI and IRU have the smallest impacts (Figure 11).

IRU increases food enterprises’ production and operation costs, so IRU mainly has negative impacts on PMFI. FU increases the production costs of food enterprises in the short term, which have negative impacts on PMFI. In the long term, it has positive impacts on PMFI by improving the financing environment and reducing financial risks. In time of high intensity of agricultural policies, the impacts of FU are greater [88]. SDU has positive impacts on PMFI through policy support and fiscal subsidy [8], and the positive impacts are more significant during periods of high liquidity.

With the continuous improvement of consumption level, people’s eating habits are constantly upgraded [73], and food varieties are upgraded accordingly to match the changes of demand [94], which leads to the fluctuation of PMFI. Therefore, CU has both negative and positive effects on PMFI.

DE and PHE first have great negative impacts on PMFI, and then have positive impacts. When DE and PHE occur, PMFI will greatly reduce through direct and indirect impacts in the short term. After food enterprises resume normal production, the profit margin will recover [88].

#### 3.3.3. The Impulse Responses of Circulation and Trade Sector

The impulse responses of food retail (FR)

The positive shocks of FU and SDU mainly have positive impacts on FR, while positive shocks of IRU, CU, DE, and PHE have both positive and negative impacts. The impacts of DE and PHE are relatively large, while the impacts of IRU are the smallest (Figure 12).

In the period of high liquidity (from April 2007 to December 2007 and from October 2009 to September 2010), due to the income effect of savings, people increased food consumption considerably, resulting in positive impacts on IRU during this period. If the money liquidity is at a moderate level, people increase savings due to the substitution effect of savings. Therefore, IRU mainly had negative impacts after 2012 [95,96].

FU increases FR by raising food prices. Meanwhile, it also stimulates food consumption by increasing residents’ income. SDU increases FR by improving residents’ income and enhancing food market construction [73]. The impacts of CU on FR are volatile. Especially after the outbreak of influenza A (H1N1) in October 2009, people significantly reduced the consumption of poultry and pork products, resulting in a significant reduction in the positive impacts of CU during this period [85].

DE and PHE rapidly increase FR in the short term by changing consumer expectations [1]; however, the food market closes due to large-scale natural disasters and highly infectious diseases, leading to a rapid decline in food retail sales [8]. Afterward, food retail sales can recover quickly due to the government policies to promote production recovery and the rapid development of e-commerce and logistics industry [8,41]. This can be used to explain why the H7N9 outbreak in December 2016 and COVID-19 in January 2020 had smaller impacts on FR than influenza A (H1N1) in October 2009.

2.The impulse responses of food products export (FPE)

The positive shocks of IRU mainly have negative impacts on FPE. SDU mainly has positive impacts, while FU, CU, DE, and PHE have both positive and negative impacts. DE and SDU have greater impacts on FPE, while CU has the least impacts (Figure 13).

IRU has negative impacts on FPE by affecting the borrowing costs of export enterprises and raising domestic food prices. After 2016, with the rise of loan interest rate and the continuous growth of agri-production material price index, the negative impacts of IRU on FPE increased. In the short term, FU can raise food prices, which causes negative impacts on FPE; in the long term, FU has positive impacts on FPE by improving the financing environment of food exporters [8,88].

SDU increases FPE by improving food quality, diversifying food supply, and improving the trade environment of food enterprises and the development prospects of food exporters [92]. Similarly, the outbreak of the Sino-US trade war led to significant decline in the positive impacts of the SDU and subsequent emergence of negative impacts. CU increases FPE rapidly by increasing the exchange rate of foreign currency to CNY in the short term [97] and has negative impacts on FPE by stimulating consumption in the long term [98].

DE and PHE first have great negative impacts on FPE and then have positive impacts. When DE and PHE occur, FPE decreases greatly in the short term; after the disasters or pandemics, FPE will experience a restorative growth [1].

3.The impulse responses of food products import (FPI)

FU mainly has positive impacts on FPI, and other external uncertainties have both positive and negative impacts. DE has the greatest impacts on FPI, followed by SDU, and IRU having the least impacts (Figure 14). In the short term, IRU has positive impacts on FPI by increasing domestic agricultural production costs. In the long run, it reduces FPI by increasing the loan costs of importing enterprises. And these negative impacts are more significant after the interest rate correction in 2016.

FU mainly increases FPI by improving the credit level and financing scale of import enterprises [88]. With the continuous growth of social economy, China’s food market is developing towards the high end; thus, SDU increases the import of high-quality food products in the short term [98]. However, with the development of food enterprises, the supply of domestic high-quality food products increases and the dependence on imports decreases. So SDU has negative impacts on FPI in the long run. The positive impacts of CU on FPI gradually decrease, indicating that the role of consumption in import increase is weakened. Moreover, there are negative impacts after the outbreak of Sino-US trade war and COVID-19.

DE and PHE first have positive impacts on FPI, and then negative impacts. The increase of FPI caused by the influenza A (H1N1) outbreak in October 2009 and H7N9 in December 2016 was significantly greater than that caused by COVID-19; considering the safety of imported food, global outbreaks of infectious diseases will strengthen quarantine on imported goods and reduce the food products import.

#### 3.3.4. The Impulse Responses of Consumer Sector

Since the sales amount in the food market equals the purchase amount of consumers (the food retail equals the food consumption), and the impacts of external uncertainties on food retail have been discussed in the circulation and trade section, this section is devoted to analyzing the impacts of external uncertainties on the food consumption price.

The impulse responses of food consumption price (FCP)

The positive shocks of FU and CU mainly have positive impacts on FCP, while IRU, SDU, DE, and PHE have both positive and negative impacts (Figure 15). IRU has positive impacts on FCP during periods of high currency liquidity (for example, April 2007 to February 2008 and May 2009 to March 2010). During periods of moderate currency liquidity, IRU has negative impacts on FCP by reducing food consumption, due to the substitution effect of savings, but these negative impacts are small [95,96]. FU has greater impacts on FCP in the time of high currency liquidity [99,100]. SDU has positive impacts on FCP by diversifying food varieties and improving quality and market construction [94]. The positive impacts of CU on FCP have a fluctuating downward trend, but the positive impacts increased significantly due to the Sino-US trade war and COVID-19.

The positive impacts of DE on FCP have great volatility. PHE has positive impacts on FCP first and then negative impacts, but both positive and negative impacts are small, reflecting the Chinese government’s effectiveness of food prices regulation.

## 4. Discussion

### 4.1. The Impacts of Interest Rate Uncertainty (IRU)

The impacts of IRU on the sales price of food enterprises and food consumption price are significantly greater than that on the purchase price of primary agri-products. On the one hand, this difference is because the food products are at the end of the supply chain, and prices are directly affected by IRU shocks, as well as the transmission of impacts that caused by IRU on other sectors. On the other hand, the price support and protection policies implemented by the Chinese government have effectively restrained the fluctuation of primary agri-products price.

IRU affects the production value of primary agri-products, production value, and sales profit of food enterprises by changing production, sales, and operating costs. In the time of high liquidity, positive shocks of IRU lead to a significant decline in the sales profit margin of food enterprises, despite the increase in its production value.aThis phenomenon causes by the sensitive responses of production costs and loan costs to changes in the interest rate.

IRU affects food retail through income effect and substitution effect of savings. The substitution effect is dominant during periods of moderate liquidity, while the income effect plays a major role during periods of high liquidity.

IRU has an impact on trade of primary agri-products and food products by affecting the agri-production material price and enterprise credit costs, and the impacts on exports are greater than that on imports. It indicates that preferential loan interest rate policy for export processing enterprises can effectively improve their resistance against risks in primary agri-products and food products exports, which is consistent with the conclusions of Qu and Kang [91] and Nordhagen et al. [88].

### 4.2. The Impacts of Financial Uncertainty (FU)

FU increases the purchase price of primary agri-products, sales price of food enterprises, and food consumption price through financialization. The impacts on the sales price and food consumption price are greater than that on the purchase price of primary agri-products. Because the purchase of agri-products is at the front of the supply chain and is less affected by risk transmission from other sectors. And price support and protection policies implemented by the Chinese government reduce the price volatility of primary agri-products, and the results verify the findings of Assefa et al. [101].

FU affects the production value of primary agri-products by changing the financing environment of primary agricultural producers and raising the price level of primary agri-products, and affects the production value and sales profit margin of food enterprises by changing the sales price, financing environment, financial risks, non-operating income, etc. FU has greater impacts on the sales profit of food enterprises. It shows that internal operation management of the enterprises reduces the risk transmitted from sales profit to production value, consistent with Kuiper and Lansink [66] and Golini et al. [102].

FU has great impacts on primary agri-products exports and small impacts on primary agri-products imports and food products trade, indicating that the relevant policies by the government and financial institutions to support the financing of agricultural export enterprises are important measures to increase agri-products exports. And this is consistent with Nordhagen et al. [88] and Zhan and Chen [8].

### 4.3. The Impacts of Social Development Uncertainty (SDU)

SDU has greater impacts on the sales price of food enterprises and food consumption than on the purchase price of primary agri-products. This is caused by the differences in risk transmission from other sectors and the implementation of agricultural protection policies. The Chinese government has implemented policies such as grain supporting procurement and agricultural production subsidies, which are of great significance to stabilize the price of agri-products. Especially from 2008 to 2013, the steady implementation of the minimum purchase price policy for wheat and rice, temporary purchase and storage policy for corn, soybean and cotton, and the substantial increase of agricultural production subsidies led to corresponding increase in the response value of agri-products purchase price during this period.

SDU has positive impacts on production value of primary agri-products, production value, and sales profit margin of food enterprises through the agricultural policies to protect primary production and improve farmers’ income (such as agricultural production subsidies, abolition of agricultural taxes, targeted poverty alleviation, and rural revitalization strategy), as well as a series of positive policies to support food enterprises (such as enterprise production subsidies, tax incentives, reducing rent and loan interest discounts), and have a greater impact on the sales profit margin. It indicates that the preferential policies implemented by the Chinese government to support the development of food enterprises have effectively reduced the internal operation costs and risks of food enterprises, and can significantly improve the profit margin. The results verify the findings of Golini et al. [102] and Nordhagen et al. [88]. Meanwhile, it can also be concluded that appropriate liquidity helps to improve effectiveness of the policies, and high intensity of agricultural policies can also promote the positive impacts of liquidity uncertainty caused by IRU and FU.

SDU increases sales by increasing consumers’ income and improving construction of food market. The purpose of improving construction of food market is to promote the diversification of consumers’ dietary structure and to meet their needs for balanced nutrition.

SDU increases the export of agri-products and food products by improving product quality, increasing varieties, and improving the trade environment, and has negative impacts on agri-products import (for example, the Sino-US trade war led to remarkable decrease of agri-products import). However, economic development also brings about upgrading of the consumer market, which increases the import of high-quality and diversified food products.

### 4.4. The Impacts of Consumption Uncertainty (CU)

CU has positive impacts on the purchase price of primary agri-products, sales price of food enterprises and food consumption price by increasing consumer confidence and expectation. The positive impacts are more significant in time of high liquidity.

Due to the necessity of agricultural production and the rigidity of food demand, the impacts of CU on production value of primary agri-products and food industry are volatile, but the overall impacts are small; these findings are consistent with Davis et al. [22].

CU first has negative impacts on the profit margin of the food industry and then positive impacts. The reason for this change is the mismatch between food supply and consumer demand caused by upgrading dietary habits. The shift from negative to positive impacts is also the process in which new products are produced and then accepted by the market, and finally growth of sales profit is achieved. This result verifies the finding of Dania et al. [32].

The impacts of CU on food retail have significant volatility. Especially when zoonotic infectious diseases occur (for example, H1N1 outbreak in October 2009), the positive impacts on food retail sales are significantly reduced. This phenomenon is affected by consumers’ attitudes toward the risk perception, which is consistent with the conclusions of Lobb et al. [15] and Seeger et al. [40].

CU increases the export of primary agri-products and food products by increasing the exchange rate of foreign currency to CNY, and increases the import of primary agri-products and food products by stimulating consumption.

### 4.5. The Impacts of Disaster Emergencies (DE)

The impacts of DE on the purchase price of primary agri-products are greater than that on the sales price and consumption price. Agricultural production has features of periodic planting and the concentrated harvest, so the occurrence of natural disasters has relatively great impacts on the price of primary agri-products. Food enterprises can reduce the transmission of domestic agri-product price fluctuations to food prices through imports, so DE has smaller impacts on the sales price of food enterprises.

With the end of the disaster and the implementation of agricultural protection policies and risk response measures, the production value of primary agri-products, the production value, and profit margin of food enterprises will gradually recover, which is consistent with Zhan and Chen [8]. Although short-duration natural disasters can significantly increase the agri-product price, they have limited impacts on the production value of primary agricultural and food industries. Long-lasting natural disasters can not only cause price increases, but also significantly decrease in the production value of primary agricultural and food industries.

DE changes consumers’ consumption expectations and increases food reserve in the short term to ensure future living needs, which lead to increased food retail. Consumption of reserved food after the disaster leads to a decline in food retail.

DE significantly reduces exports and increases imports of agri-products and food products in the short term. After the disaster, exports of agri-products and food products experience restorative growth, and their imports declined significantly. With the continuous improvement of agriculture-related policies, agri-products trade (especially imports) is gradually affected by policies, while the impacts of DE are gradually reduced.

### 4.6. The Impacts of Public Health Emergencies (PHE)

The impacts of PHE on the purchase price of primary agri-products are smaller than that on sales price and food consumption price, which is consistent with Nordhagen et al. [88]. This diversity is also caused by the differences in risk transmission from other sectors and the implementation of agriculture protection policies. Meanwhile, occurrence of zoonotic infectious diseases (such as malaria, anthrax, plague, H5N1 in 2005, H1N1 in 2009, and H7N9 in 2017) leads to a significant decline in the positive impacts of PHE on the price of agricultural products, by changing consumer decision making. This is consistent with Dhand et al. [85].

PHE has relatively small impacts on the production value of primary agri-products, mainly due to the series of agricultural support and protection policies, as well as the emergency prevention and control measures. PHE first has great negative impacts on the production value and profits of the food enterprises, and then have positive impacts. Especially after the occurrence of zoonotic infectious diseases, the negative impacts on the production value and profit margin increase, and the positive impacts decrease. After the pandemic, the production value recovers rapidly, but sales profit margin recovers slightly, which may lead to the long-term loss of enterprises, and it is consistent with Nordhagen et al. [88]. Therefore, support and protection policies for food enterprises practice by the government after the pandemic help food enterprises to resume normal operation.

The impacts of PHE on food retail sales are much greater than that on the production value, price and profit margin of the food industry, reflecting that China’s food supply chain has a certain resilience, and this result verifies the findings of Fan et al. [90]. Meanwhile, the continuous improvement of agricultural policies and emergency measures, and the rapid development of e-commerce and logistics have gradually reduced the impact of PHE on food retail. Therefore, after the outbreak of COVID-19 in China, the food supply in Wuhan, where the most serious pandemic was still sufficient, and the food supply chain had not been disrupted for a long time. This can be attributed to the efforts of the Chinese government in ensuring the production, storage, and transportation of agri-products [8].

PHE significantly reduces export and increases imports of agri-products and food products in the short term, but export of agri-products and food products experiences restorative growth after the pandemic. Import of food products can be reduced during the global infectious diseases (such as COVID-19 in 2020), and the terms of trade will be more stringent (such as strengthening quarantine and screening import markets), because of concerns over the safety of imported goods.

## 5. Conclusions and Limitations

### 5.1. Conclusions

This study divided the main sectors of the food supply chain and fully considered the time-varying quality, complexity, and high-frequency occurrence of external uncertainties as the entry points, then selected a large number of monthly indicators to measure the external uncertainties, and used three-dimensional impulse responses based on the TVP-FAVAR-SV model to depict the dynamic impacts of external uncertainties on food supply chain. The impacts of combination of various uncertainty elements and the transmission of these impacts among various sectors of the food supply chain are the focus of this study. It was found that the impacts of all external uncertainty elements were significantly different: social and natural public emergencies had great impacts on aspects of retail sales, profit margin, and international trade, while economic uncertainties had great impacts on aspects of price, production, and international trade. It was also found that the combination of different external uncertainty elements aggravated or reduced the risks of food supply chain. With the continuous improvement of the agricultural policy system, the negative impacts of economic uncertainties on the food supply chain were gradually diminished, and the appropriate liquidity improved the positive impacts of relevant policies. The integrated emergency measures and complete market systems reduced the negative impacts of disaster emergencies and public health emergencies on the production sector, and shortened the time for food enterprises to resume operation and end the profit losses. Unlike risks, some uncertainty elements had both positive and negative impacts in the whole sample period. The magnitude and direction of the impacts of various uncertainties in different periods had time-varying characteristics, and the time-varying characteristics were caused by the impacts of combination of various uncertainty elements and the transmission of these impacts.

### 5.2. Limitations

This paper recognizes and acknowledges that some limitations are still in variable selection and content analysis when analyzing the dynamic impacts of external uncertainties on the food supply chain, which provides approaches for future research. When identifying the external uncertainties of the food system, the paper tried to summarize the socio-economic environment and natural environment, and finally selected many indicators from public health emergencies, disaster emergencies, and economic uncertainties to measure the complexity of external uncertainties. However, the uncertainty risks of the food system are multifaceted, and especially in recent years, the pressure of resources, environment, and energy consumption has seriously challenged the sustainable stability of the food system. This paper is short of risk analysis from the aspects of resources, environment, and energy. Moreover, the Chinese government has implemented a series of policies and measures to achieve the carbon peak and neutrality goals. These policies and measures also have long-term and important impacts on the sustainable development of the agri-food system. However, this paper is short of the uncertainty analysis of relevant policies and measures. ITo comprehensively analyze the impacts of external uncertainties to ensure the sustainable stability of the food system, future research could focus on: (1) selecting impact factors from energy and environment as important indicators and summarizing them into the external uncertainty system, such as energy prices, production, trade, as well as carbon emissions, land use, agricultural material consumption, and other elements; (2) using a variety of quantitative methods to study the impacts of relevant policies and measures implemented by the Chinese government on the food system, such as data crawling technology, event and text analysis method, etc., to achieve multi-scale research on the sustainable security of the food supply chain.

## Figures and Tables

**Figure 1 foods-11-02552-f001:**
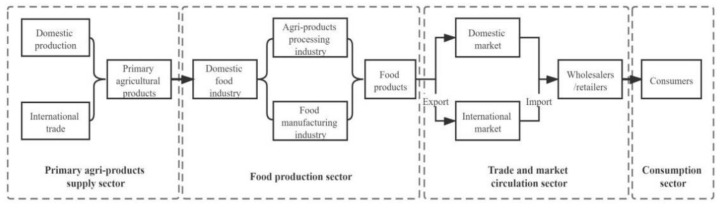
The major sectors of the food supply chain.

**Figure 2 foods-11-02552-f002:**
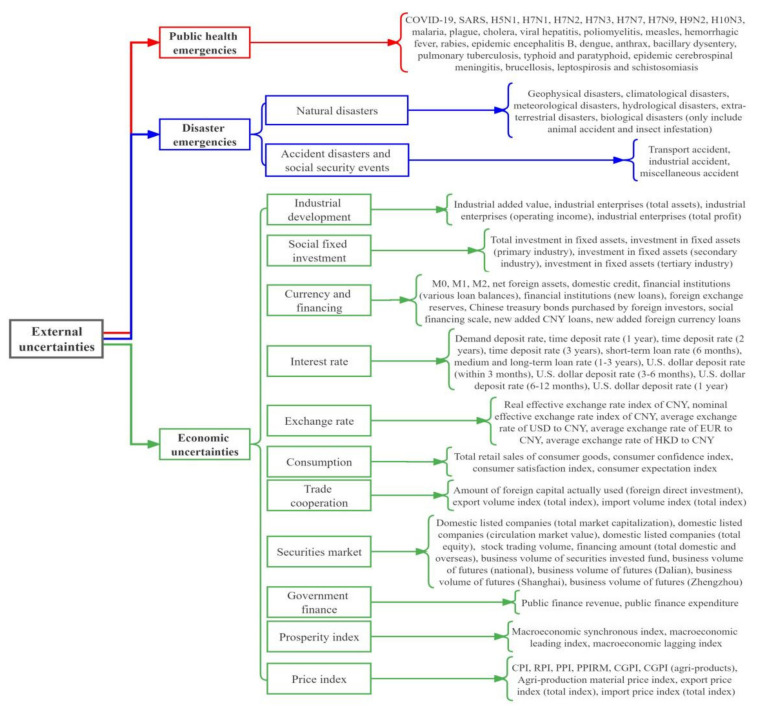
Sources and classification of external uncertainties.

**Figure 3 foods-11-02552-f003:**
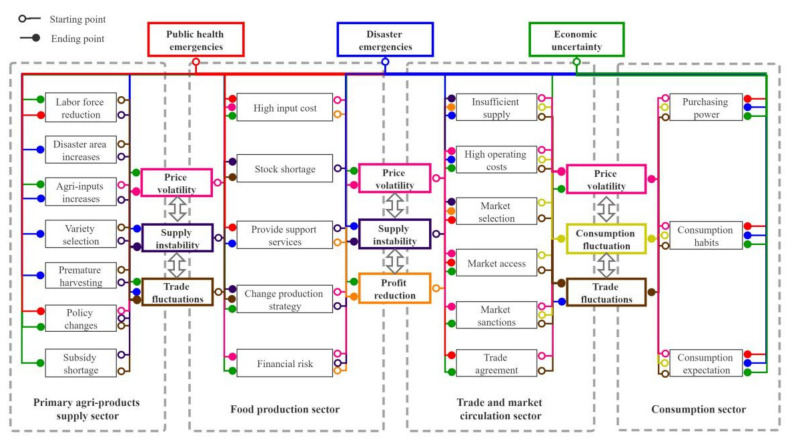
Main impacts caused by external uncertainties in food supply chain. Note: The hollow points in Figure 3 represent the starting points of impacts transmission, and the solid points represent the end points of impacts transmission; the text boxes of different colors represent different impact sources and impact results, and the solid lines of the same color represent the transmission process of the same impact.

**Figure 4 foods-11-02552-f004:**
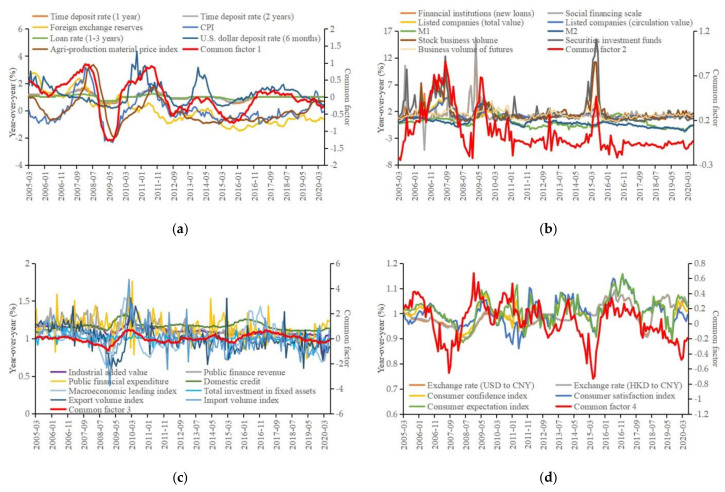
Four common factors of economic uncertainties. (**a**) Common factor 1 and related variables. (**b**) Common factor 2 and related variables. (**c**) Common factor 3 and related variables. (**d**) Common factor 4 and related variables. Note: In order to describe the trend between common factors and related variables clearly, some variables related to each common factor are selected as representatives for display.

**Figure 5 foods-11-02552-f005:**
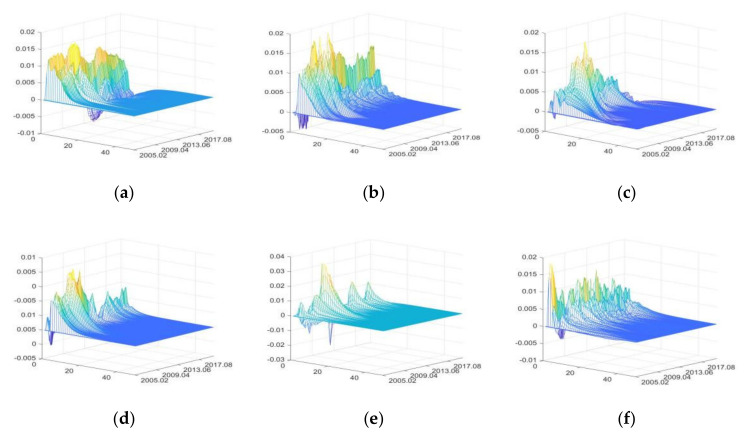
The three-dimensional impulse responses of PAPP to positive shocks of external uncertainties. (**a**) Shocks of IRU. (**b**) Shocks of FU. (**c**) Shocks of SDU. (**d**) Shocks of CU. (**e**) Shocks of DE. (**f**) Shocks of PHE. Note: The color is automatically generated by MATLAB software. The yellow color represents high value, blue color represents low value, and green color represents intermediate value.

**Figure 6 foods-11-02552-f006:**
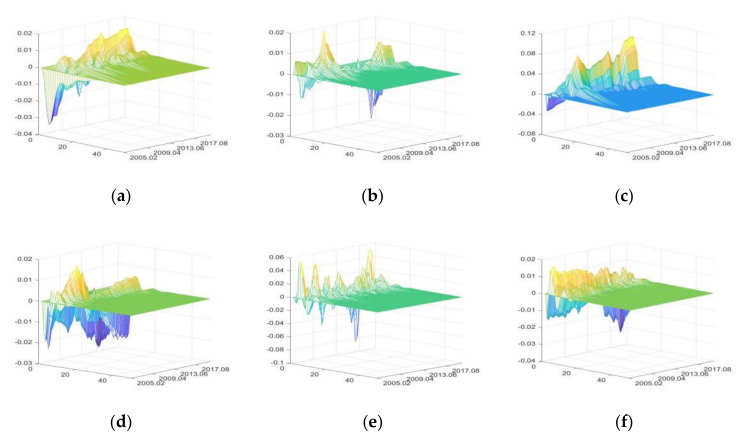
The three-dimensional impulse responses of PVPA to positive shocks of external uncertainties. (**a**) Shocks of IRU. (**b**) Shocks of FU. (**c**) Shocks of SDU. (**d**) Shocks of CU. (**e**) Shocks of DE. (**f**) Shocks of PHE.

**Figure 7 foods-11-02552-f007:**
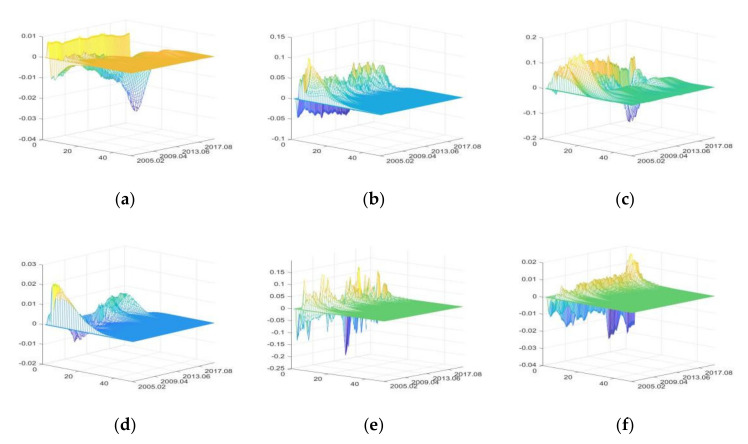
The three-dimensional impulse responses of PAE to positive shocks of external uncertainties. (**a**) Shocks of IRU. (**b**) Shocks of FU. (**c**) Shocks of SDU. (**d**) Shocks of CU. (**e**) Shocks of DE. (**f**) Shocks of PHE.

**Figure 8 foods-11-02552-f008:**
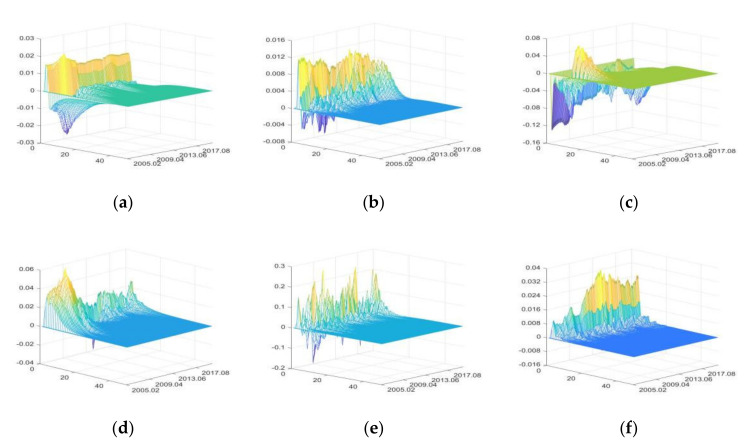
The three-dimensional impulse responses of PAI to positive shocks of external uncertainties. (**a**) Shocks of IRU. (**b**) Shocks of FU. (**c**) Shocks of SDU. (**d**) Shocks of CU. (**e**) Shocks of DE. (**f**) Shocks of PHE.

**Figure 9 foods-11-02552-f009:**
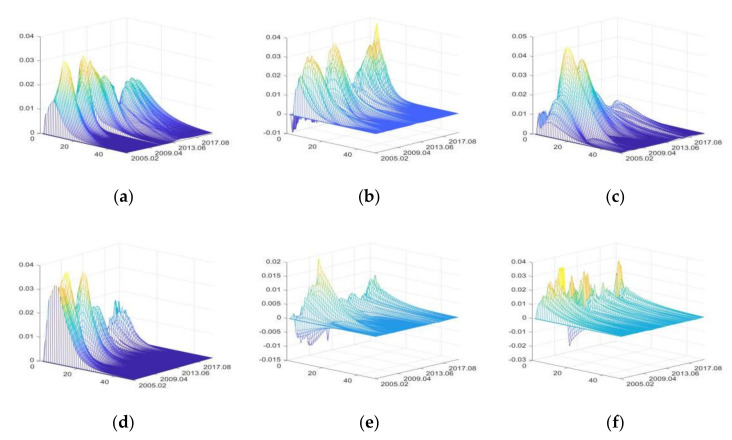
The three-dimensional impulse responses of SPFE to positive shocks of external uncertainties. (**a**) Shocks of IRU. (**b**) Shocks of FU. (**c**) Shocks of SDU. (**d**) Shocks of CU. (**e**) Shocks of DE. (**f**) Shocks of PHE.

**Figure 10 foods-11-02552-f010:**
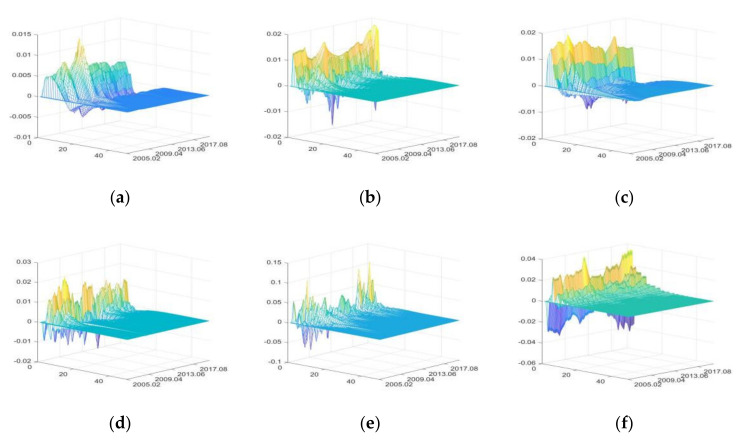
The three-dimensional impulse responses of PVFI to positive shocks of external uncertainties. (**a**) Shocks of IRU. (**b**) Shocks of FU. (**c**) Shocks of SDU. (**d**) Shocks of CU. (**e**) Shocks of DE. (**f**) Shocks of PHE.

**Figure 11 foods-11-02552-f011:**
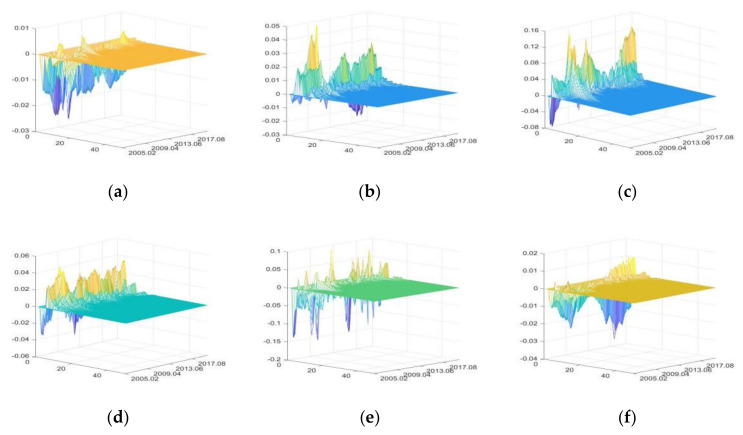
The three-dimensional impulse responses of PMFI to positive shocks of external uncertainties. (**a**) Shocks of IRU. (**b**) Shocks of FU. (**c**) Shocks of SDU. (**d**) Shocks of CU. (**e**) Shocks of DE. (**f**) Shocks of PHE.

**Figure 12 foods-11-02552-f012:**
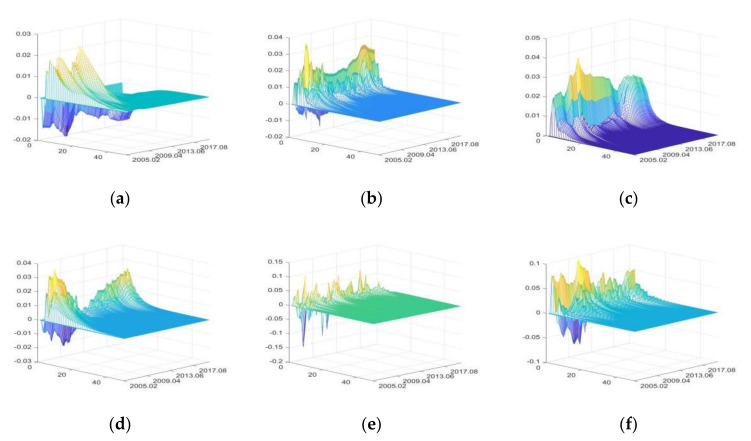
The three-dimensional impulse responses of FR to positive shocks of external uncertainties. (**a**) Shocks of IRU. (**b**) Shocks of FU. (**c**) Shocks of SDU. (**d**) Shocks of CU. (**e**) Shocks of DE. (**f**) Shocks of PHE.

**Figure 13 foods-11-02552-f013:**
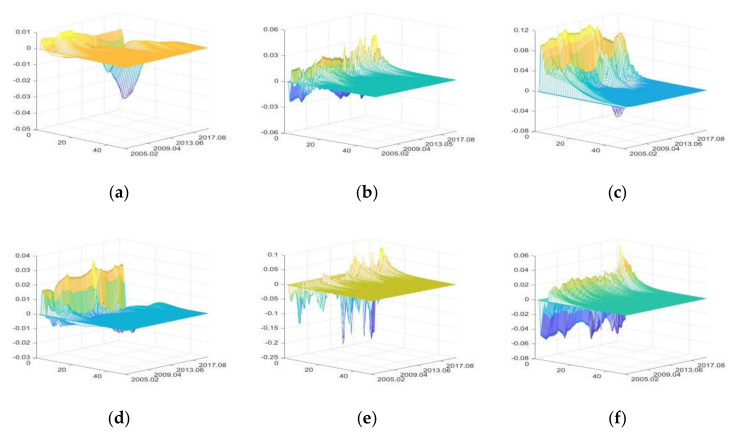
The three-dimensional impulse responses of FPE to positive shocks of external uncertainties. (**a**) Shocks of IRU. (**b**) Shocks of FU. (**c**) Shocks of SDU. (**d**) Shocks of CU. (**e**) Shocks of DE. (**f**) Shocks of PHE.

**Figure 14 foods-11-02552-f014:**
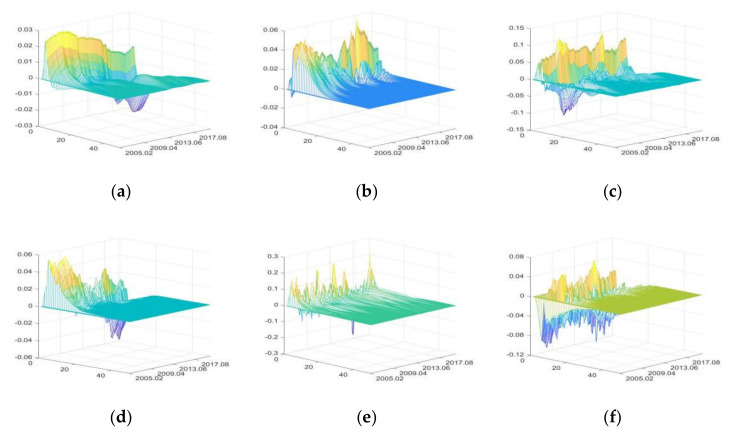
The three-dimensional impulse responses of FPI to positive shocks of external uncertainties. (**a**) Shocks of IRU. (**b**) Shocks of FU. (**c**) Shocks of SDU. (**d**) Shocks of CU. (**e**) Shocks of DE. (**f**) Shocks of PHE.

**Figure 15 foods-11-02552-f015:**
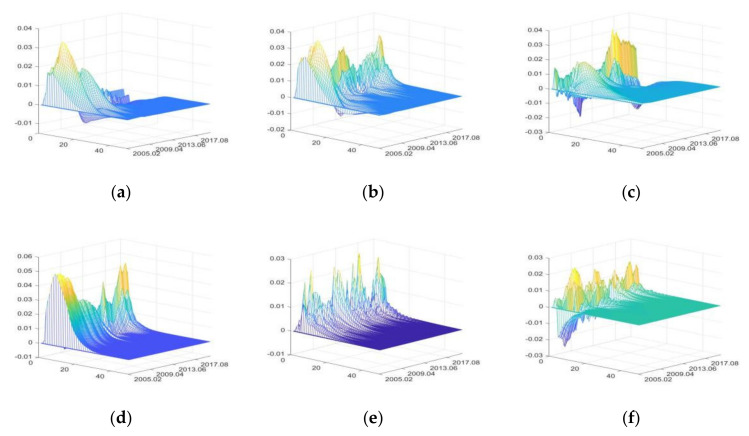
The three-dimensional impulse responses of FCP to positive shocks of external uncertainties. (**a**) Shocks of IRU. (**b**) Shocks of FU. (**c**) Shocks of SDU. (**d**) Shocks of CU. (**e**) Shocks of DE. (**f**) Shocks of PHE.

**Table 1 foods-11-02552-t001:** Descriptive statistics of data.

Variables	Units	Mean	Max	Min	Std. Dev.	Data Source
Primary agri-products purchase price	%	103.164	119.800	94.200	5.282	Data from the Chinese National Bureau of Statistics
Sales price of food enterprise	%	102.383	114.000	96.300	3.547	
Food consumption price	%	105.763	123.300	95.600	5.739	
Production value of primary agri-products	%	3554.244	9234.000	338.909	2119.046	
Production value of food industry	10^8^ CNY	6667.228	12,616.400	1332.065	2975.821	
Profit margin of food industry	%	5.365	11.014	3.236	1.215	
Food retail	10^8^ CNY	763.172	1902.720	137.600	466.990	
Primary agri-products export	10^8^ CNY	45.507	97.041	25.690	14.337	Data from the China Customs Database
Primary agri-products import	10^8^ CNY	203.113	419.828	36.411	92.870	
Food products export	10^8^ CNY	33.555	67.762	10.560	12.783	
Food products import	10^8^ CNY	33.278	81.089	5.680	17.919	
Public health emergencies	person	1296.661	3755.000	370.000	510.675	Data from the NHC of the People’s Republic of China
Disaster emergencies	10^4^ person	610.0386	14,011.69	0.000	1663.497	Data from the EM-DAT Database
Industrial added value	%	10.391	29.200	−25.867	5.970	Data from the Wind Database
Public finance revenue	10^8^ CNY	6979.638	16,055.720	1928.032	3014.549	
Public finance expenditure	10^8^ CNY	7903.190	21,132.330	1388.390	4582.977	
Total investment in fixed assets	10^8^ CNY	22,340.870	54,355.990	2962.435	13,122.510	
Investment in fixed assets (primary industry)	10^8^ CNY	569.680	2025.149	2.955	487.782	
Investment in fixed assets (secondary industry)	10^8^ CNY	8776.423	20,283.590	976.327	4908.527	
Investment in fixed assets (tertiary industry)	10^8^ CNY	12,995.440	35,073.990	1870.539	8027.910	
Total retail sales of consumer goods	10^8^ CNY	17,946.700	38,776.700	4663.300	9814.165	
M0	10^8^ CNY	37,630.290	59,745.760	19,543.270	10,163.590	
M1	10^8^ CNY	227,324.900	383,947.900	87,493.670	91,109.360	
M2	10^8^ CNY	738,079.400	1,356,420.000	244,487.800	340,520.100	
Net foreign assets	10^8^ CNY	217,014.400	294,660.200	56,902.350	71,223.260	
Domestic credit	10^8^ CNY	962,760.900	2,351,762.000	226,979.400	638,215.500	
Financial institutions (various loan balances)	10^8^ CNY	706,143.300	1,651,999.000	181,083.000	425,656.300	
Financial institutions (new loans)	10^8^ CNY	5861.137	21,587.960	158.835	3697.832	
Foreign exchange reserves	10^8^ CNY	131,456.300	178,312.700	49,082.990	34,057.340	
Treasury bonds purchased by foreign investors	10^8^ CNY	966.227	2745.323	188.598	475.382	
Social financing scale	10^8^ CNY	9388.996	32,997.260	2.771	5890.928	
New added CNY loan	10^8^ CNY	7976.525	35,668.360	−314.000	5762.006	
New added foreign currency loan	10^8^ CNY	177.642	2542.000	−2344.000	645.705	
Real effective exchange rate index of CNY	%	108.644	130.930	82.390	14.853	
Nominal effective exchange rate index of CNY	%	106.466	126.540	83.950	11.929	
Average exchange rate of USD to CNY	—	6.844	8.277	6.104	0.610	
Average exchange rate of EUR to CNY	—	8.655	11.037	6.626	1.168	
Average exchange rate of HKD to CNY	—	0.880	1.064	0.787	0.078	
Demand deposit rate	%	0.457	0.810	0.350	0.162	
Time deposit rate (1 year)	%	2.424	4.140	1.500	0.798	
Time deposit rate (2 year)	%	3.050	4.680	2.100	0.854	
Time deposit rate (3year)	%	3.652	5.400	2.750	0.849	
Short-term loan rate (6 months)	%	5.169	6.570	4.350	0.691	
Medium and long-term loan rate (1–3 years)	%	5.714	7.560	4.750	0.832	
USD deposit rate: within 3 months	%	2.058	4.811	0.320	1.252	
USD deposit rate (3–6 months)	%	2.598	5.400	0.570	1.323	
USD deposit rate (6–12 months)	%	2.814	5.890	0.740	1.245	
USD deposit rate (1 year)	%	3.086	7.110	1.180	1.256	
Foreign direct investment	10^8^ CNY	89.959	187.800	38.700	29.336	
Export volume index (total index)	%	109.364	154.200	76.100	14.028	
Import volume index (total index)	%	107.976	163.500	63.700	11.801	
Export price index (total index)	%	102.281	111.900	90.700	4.576	
Import price index (total index)	%	102.364	122.700	79.600	9.471	
Listed companies (total market capitalization)	10^8^ CNY	222,420.000	446,901.700	29,289.820	112,153.500	
Listed companies (circulating market value)	10^8^ CNY	160,237.200	412,054.600	9156.037	103,610.100	
Domestic listed companies (total equity)	10^8^ CNY	38,257.570	71,852.640	7151.950	18,709.630	
Stock business volume	10^8^ CNY	46,140.270	258,418.300	1371.614	41,122.770	
Stock trading volume	10^8^ shares	5400.305	20,462.790	305.410	4294.477	
Financing amount (total domestic and overseas)	10^8^ CNY	1065.552	4157.350	0.763	966.994	
Business volume of securities invested fund	10^8^ CNY	724.977	10,080.260	13.341	1300.519	
Business volume of futures (national)	10^8^ CNY	120,079.000	694,460.100	5028.965	99,941.170	
Business volume of futures (Dalian)	10^8^ CNY	22,961.500	64,644.090	1268.152	13,234.480	
Business volume of futures (Shanghai)	10^8^ CNY	37,594.110	106,166.400	2663.333	21,873.080	
Business volume of futures (Zhengzhou)	10^8^ CNY	15,202.100	97,506.910	1097.479	11,964.630	
Macroeconomic synchronous index	%	99.410	104.700	82.685	3.874	
Macroeconomic leading index	%	101.428	105.900	95.564	2.153	
Macroeconomic lagging index	%	97.165	103.000	89.500	3.055	
Consumer confidence index	%	109.125	126.600	97.000	7.409	
Consumer satisfaction index	%	105.578	121.000	90.000	7.567	
Consumer expectation index	%	111.469	130.700	99.000	7.810	
CPI	%	2.659	8.700	−1.800	1.908	
RPI	%	1.921	8.100	−2.500	2.007	
PPI	%	1.334	10.060	−8.200	4.362	
PPIRM	%	2.337	15.390	−11.680	6.248	
CGPI	%	101.457	110.300	92.000	4.837	
CGPI (agri-products)	%	104.588	120.200	94.400	6.221	
Agri-production material price index	%	4.452	24.800	−7.513	6.073	

Note: In the primary agri-products supply sector, the purchase price index of agri-products is adopted as a substitute for the purchase price of primary agri-products. In the production sector, the producer price index of food enterprises is used as a substitute for the sales price of food enterprises. In the circulation and trade sector, the retail sales amount of grain, oil, and food is used as a substitute for the food retail. In the consumption sector, consumer price index of food is adopted as a substitute for the food consumption price.

**Table 2 foods-11-02552-t002:** Unit root test.

Variable Name	Symbol	Test Type (C,T,L)	ADF-Statistic	1% Critical-Value	5% Critical-Value	*p*-Value
Primary agri-products purchase price	PAPP	(C,0,4)	−3.7879	−3.4666	−2.8774	0.0036 ***
Sales price of food enterprise	SPFE	(C,0,4)	−3.5953	−3.4666	−2.8774	0.0067 ***
Food consumption price	FCP	(C,0,4)	−3.0987	−3.4666	−2.8774	0.0284 **
Primary agri-products export	PAE	(C,0,4)	−3.8366	−3.4665	−2.8773	0.0031 ***
Primary agri-products import	PAI	(C,0,4)	−3.5978	−3.4665	−2.8773	0.0067 ***
Production value of primary agri-products	PVPA	(C,T,0)	−5.2568	−4.0084	−3.4343	0.0001 ***
Production value of food industry	PVFI	(C,T,0)	−5.6640	−4.0084	−3.4343	0.0000 ***
Profit margin of food industry	PMFI	(C,T,0)	−8.1105	−4.0084	−3.4343	0.0000 ***
Food retail	FR	(C,T,0)	−8.4175	−4.0084	−3.4343	0.0000 ***
Food products export	FPE	(C,T,0)	−5.2731	−4.0084	−3.4343	0.0001 ***
Food products import	FPI	(C,T,0)	−5.0751	−4.0084	−3.4343	0.0002 ***
Public health emergencies	PHE	(C,T,0)	−10.0659	−4.0084	−3.4343	0.0000 ***
Disaster emergencies	DE	(C,T,0)	−12.7806	−4.0084	−3.4343	0.0000 ***
Interest rate uncertainty factor	IRU	(0,0,4)	−3.9302	−2.5779	−1.9426	0.0001 ***
Financial uncertainty factor	FU	(0,0,4)	−2.5491	−2.5779	−1.9426	0.0108 **
Social development uncertainty factor	SDU	(0,0,4)	−3.4806	−2.5779	−1.9426	0.0006 ***
Consumption uncertainty factor	CU	(0,0,4)	−3.1611	−2.5779	−1.9426	0.0017 ***

Note: In the test type, C, T, and L denote the intercept, trend, and lag periods in ADF test, respectively; ** Significant at the 0.05 level, *** Significant at the 0.01 level.

## Data Availability

The datasets used and analyzed during the current study are available from the corresponding author on reasonable request.

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
