# Peer review of "Dynamic Impacts of External Uncertainties on the Stability of the Food Supply Chain: Evidence from China"

_foods, 2022, doi:10.3390/foods11172552_

Round 1

Reviewer 1 Report

Dear Authors,

thanks for this interesting proposal entitled "Dynamic impacts of external uncertainties on the stability of food supply chain: Evidence from China”. At this stage the paper requires minor revision due to some related to the both structural and content aspects. Above some suggestions.

 In relation to the structural aspects, those are related to the conclusions section. This reviewer suggests to create in the section Conclusion, at the end of it, the limitation and future research part. 

In relation to the content aspects, following some suggestions.

Firstly, the Abstract requires few actions but really important. This section must briefly present the purpose, methodology, originality and findings of the research. In this important part this reviewer suggests to provide just a sentence referred to the methodological aspects and to the results (all the statistical findings ca be read in the full paper in the specific section).

Secondly, literature needs to be improved. It is strongly suggested to discuss about the food and supply chain (that are a central points of your proposal). Among other literature, the following references are strongly suggested to improve the “Introduction section” and also better qualify the research problems:

-          Cillo, V., Gavinelli, L., Ceruti, F., Perano, M., Solima, L., 2019, A sensory perspective in the Italian beer market, British Food Journal, 121, 9, 2036-2051.

-          Pellicano, M., Ciasullo, M. V., Festa, G., 2015, January, The analysis of the relational context in wine tourism. In Proceedings of the 1st Euromed Specialized Niche Conference on “Contemporary Trends and Perspectives in Wine and Agrifood Management”, University of Salento, Lecce, 16-17 January (pp. 307-328);-    

-          Mital, M., Del Giudice, M., Papa, A., 2018, Comparing supply chain risks for multiple product categories with cognitive mapping and Analytic Hierarchy Process, Technological Forecasting and Social Change, 131, 159-170.

Thirdly, methodology section: the research design and process need to be generally introduced and then better described specifically in the specific section. This reviewer suggests to start this section with a general introduction through which describe step by step your research design and process and then going in a specific description in the specific sections. More precisely, there isn’t a general section in which explain the research design, your approach (i.e. qualitative, quantitative or mix method) and why you are choose this methodology.

Finally, a proofread can also improve the quality of this interesting proposal.

Good luck with your paper.

Author Response

Dear reviewers,

We would like to express our deep gratitude for your comments. We made point-to-point revisions according to the review comments. Please check the attached file for details.

Thank you very much for your consideration.

Best regards!

Reviewer 2 Report

Authors have deliberated the impact of various social, economic and environmental uncertainties on the food supply chain. Furthermore, they have narrowed it to disaster, public health emergencies and economic uncertainties and corresponding indicators are clustered. In addition to the same some comments to the authors are as follows:

Comment 1: Authors needs to detail the respondent profile chosen for the data collection in Sub-section 2.5 named Data Sources.

Comment 2: Authors are requested to mention the name of the software/support system used for carrying out the distinctive empirical analysis respectively.

Comment 3: Authors claim the timeline of observation from 2005 to 2020. But in the results supporting evidence is provided from the year of 2008 to validate the instances under consideration.

Comment 4: In the food supply chains customer demand remains the primary. This study needs to incorporate some consumer perspectives in the results. It will improve the dimensions of results and will strengthen the demand patterns analysis in food supply chain.

Author Response

(The authors gave the same response as above.)
